# Teaching Inverse Reinforcement Learners via Features and Demonstrations

**Luis Haug**
Department of Computer Science
ETH Zurich
lhaug@inf.ethz.ch

**Sebastian Tschiatschek**
Microsoft Research
Cambridge, UK
setschia@microsoft.com

**Adish Singla**
Max Planck Institute for Software Systems
Saarbrücken, Germany
adishs@mpi-sws.org

## Abstract

Learning near-optimal behaviour from an expert's demonstrations typically relies on the assumption that the learner knows the features that the true reward function depends on. In this paper, we study the problem of learning from demonstrations in the setting where this is *not* the case, i.e., where there is a mismatch between the worldviews of the learner and the expert. We introduce a natural quantity, the *teaching risk*, which measures the potential suboptimality of policies that look optimal to the learner in this setting. We show that bounds on the teaching risk guarantee that the learner is able to find a near-optimal policy using standard algorithms based on inverse reinforcement learning. Based on these findings, we suggest a teaching scheme in which the expert can decrease the teaching risk by updating the learner's worldview, and thus ultimately enable her to find a near-optimal policy.

## 1 Introduction

Reinforcement learning has recently led to impressive and widely recognized results in several challenging application domains, including game-play, e.g., of classical games (Go) and Atari games. In these applications, a clearly defined reward function, i.e., whether a game is won or lost in the case of Go or the number of achieved points in the case of Atari games, is optimized by a reinforcement learning agent interacting with the environment.

However, in many applications it is very difficult to specify a reward function that captures all important aspects. For instance, in an autonomous driving application, the reward function of an autonomous vehicle should capture many different desiderata, including the time to reach a specified goal, safe driving characteristics, etc. In such situations, learning from demonstrations can be a remedy, transforming the need of specifying the reward function to the task of providing an expert's demonstrations of desired behaviour; we will refer to this expert as the *teacher*. Based on these demonstrations, a learning agent, or simply *learner* attempts to infer a (stochastic) policy that approximates the feature counts of the teacher's demonstrations. Examples for algorithms that can be used to that end are those in [Abbeel and Ng, 2004] and [Ziebart et al., 2008], which use inverse reinforcement learning (IRL) to estimate a reward function for which the demonstrated behaviour is optimal, and then derive a policy based on that.

For this strategy to be successful, i.e., for the learner to find a policy that achieves good performance with respect to the reward function set by the teacher, the learner has to know what features the teacher

considers and the reward function depends on. However, as we argue, this assumption does not hold in many real-world applications. For instance, in the autonomous driving application, the teacher, e.g., a human driver, might consider very different features, including high-level semantic features, while the learner, i.e., the autonomous car, only has sensory inputs in the form of distance sensors and cameras providing low-level semantic features. In such a case, there is a mismatch between the teacher's and the learner's features which can lead to degraded performance and unexpected behaviour of the learner.

In this paper we investigate exactly this setting. We assume that the true reward function is a linear combination of a set of features known to the teacher. The learner also assumes that the reward function is linear, but in features which are different from the truly relevant ones; e.g., the learner could only observe a subset of those features. In this setting, we study the potential decrease in performance of the learner as a function of the learner's worldview. We introduce a natural and easily computable quantity, the *teaching risk*, which bounds the maximum possible performance gap of the teacher and the learner.

We continue our investigation by considering a teaching scenario in which the teacher can provide additional features to the learner, e.g., add additional sensors to an autonomous vehicle. This naturally raises the question which features should be provided to the learner to maximize her performance. To this end, we propose an algorithm that greedily minimizes the teaching risk, thereby shrinking the maximal gap in performance that policies optimized with respect to the learner's resp. teacher's worldview can have.

Or main contributions are:

1. We formalize the problem of worldview mismatch for reward computation and policy optimization based on demonstrations.

2. We introduce the concept of *teaching risk*, bounding the maximal performance gap of the teacher and the learner as a function of the learner's worldview and the true reward function.

3. We formally analyze the teaching risk and its properties, giving rise to an algorithm for teaching a learner with an incomplete worldview.

4. We substantiate our findings in a large set of experiments.

## 2  Related Work

Our work is related to the area of algorithmic machine teaching, where the objective is to design effective teaching algorithms to improve the learning process of a learner [Zhu et al., 2018, Zhu, 2015]. Machine teaching has recently been studied in the context of diverse real-world applications such as personalized education and intelligent tutoring systems [Hunziker et al., 2018, Rafferty et al., 2016, Patil et al., 2014], social robotics [Cakmak and Thomaz, 2014], adversarial machine learning [Mei and Zhu, 2015], program synthesis [Mayer et al., 2017], and human-in-the-loop crowdsourcing systems [Singla et al., 2014, Singla et al., 2013]. However, different from ours, most of the current work in machine teaching is limited to supervised learning settings, and to a setting where the teacher has full knowledge about the learner's model.

Going beyond supervised learning, [Cakmak et al., 2012, Brown and Niekum, 2018, Kamalaruban et al., 2018] have studied the problem of teaching an IRL agent, similar in spirit to what we do in our work. Our work differs from their work in several aspects—they assume that the teacher has full knowledge of the learner's feature space, and then provides a near-optimal set/sequence of demonstrations; we consider a more realistic setting where there is a mismatch between the teacher's and the learner's feature space. Furthermore, in our setting, the teaching signal is a mixture of demonstrations and features.

Our work is also related to teaching via explanations and features as explored recently by [Aodha et al., 2018] in a supervised learning setting. However, we explore the space of teaching by explanations when teaching an IRL agent, which makes it technically very different from [Aodha et al., 2018]. Another important aspect of our teaching algorithm is that it is adaptive in nature, in the sense that the next teaching signal accounts for the current performance of the learner (i.e., worldview in our setting). Recent work of [Chen et al., 2018, Liu et al., 2017, Yeo et al., 2019] have studied adaptive teaching algorithms, however only in a supervised learning setting.

Apart from machine teaching, our work is related to [Stadie et al., 2017] and [Sermanet et al., 2018], which also study imitation learning problems in which the teacher and the learner view the world differently. However, these two works are technically very different from ours, as we consider the problem of providing teaching signals under worldview mismatch from the perspective of the teacher.

## 3   The Model

**Basic definitions.**   Our environment is described by a *Markov decision process* $\mathcal{M} = (S, A, T, D, R, \gamma)$, where $S$ is a finite set of states, $A$ is a finite set of available actions, $T$ is a family of distributions on $S$ indexed by $S \times A$ with $T_{s,a}(s')$ describing the probability of transitioning from state $s$ to state $s'$ when action $a$ is taken, $D$ is the initial-state distribution on $S$ describing the probability of starting in a given state, $R\colon S \to \mathbb{R}$ is a reward function and $\gamma \in (0,1)$ is a discount factor. We assume that there exists a feature map $\phi\colon S \to \mathbb{R}^k$ such that the reward function is linear in the features given by $\phi$, i.e.,

$$R(s) = \langle \mathbf{w}^*, \phi(s) \rangle$$

for some $\mathbf{w}^* \in \mathbb{R}^k$ which we assume to satisfy $\|\mathbf{w}^*\| = 1$.

By a *policy* we mean a family of distributions on $A$ indexed by $S$, where $\pi_s(a)$ describes the probability of taking action $a$ in state $s$. We denote by $\Pi$ the set of all such policies. The performance measure for policies we are interested in is the *expected discounted reward* $R(\pi) := \mathbb{E}\left(\sum_{t=0}^{\infty} \gamma^t R(s_t)\right)$, where the expectation is taken with respect to the distribution over trajectories $(s_0, s_1, \dots)$ induced by $\pi$ together with the transition probabilities $T$ and the initial-state distribution $D$. We call a policy $\pi$ *optimal* for the reward function $R$ if $\pi \in \arg\max_{\pi' \in \Pi} R(\pi')$. Note that

$$R(\pi) = \langle \mathbf{w}^*, \mu(\pi) \rangle,$$

where $\mu\colon \Pi \to \mathbb{R}^k$, $\pi \mapsto \mathbb{E}\left(\sum_{t=0}^{\infty} \gamma^t \phi(s_t)\right)$, is the map taking a policy to its vector of *(discounted) feature expectations*. Note also that the image $\mu(\Pi)$ of this map is a bounded subset of $\mathbb{R}^k$ due to the finiteness of $S$ and the presence of the discounting factor $\gamma \in (0,1)$; we denote by $\operatorname{diam} \mu(\Pi) = \sup_{\mu_0, \mu_1 \in \mu(\Pi)} \|\mu_0 - \mu_1\|$ its diameter. Here and in what follows, $\|\cdot\|$ denotes the Euclidean norm.

**Problem formulation.**   We consider a learner $L$ and a teacher $T$, whose ultimate objective is that $L$ finds a near-optimal policy $\pi^L$ with the help of $T$.

The challenge we address in this paper is that of achieving this objective under the assumption that there is a *mismatch between the worldviews of $L$ and $T$*, by which we mean the following: Instead of the "true" feature vectors $\phi(s)$, $L$ observes feature vectors $A^L \phi(s) \in \mathbb{R}^\ell$, where

$$A^L\colon \mathbb{R}^k \to \mathbb{R}^\ell$$

is a linear map (i.e., a matrix) that we interpret as $L$'s worldview. The simplest case is that $A^L$ selects a subset of the features given by $\phi(s)$, thus modelling the situation where $L$ only has access to a subset of the features relevant for the true reward, which is a reasonable assumption for many real-world situations. More generally, $A^L$ could encode different weightings of those features.

The question we ask is whether and how $T$ can provide demonstrations or perform other teaching interventions, in a way such as to make sure that $L$ achieves the goal of finding a policy with near-optimal performance.

**Assumptions on the teacher and on the learner.**   We assume that $T$ knows the full specification of the MDP as well as $L$'s worldview $A^L$, and that she can help $L$ to learn in two different ways:

1. By providing $L$ with demonstrations of behaviour in the MDP;
2. By updating $L$'s worldview $A^L$.

Demonstrations can be provided in the form of trajectories sampled from a (not necessarily optimal) policy $\pi^T$, or in the form of (discounted) feature expectations of such a policy. The method by which $T$ can update $A^L$ will be discussed in Section 5. Based on $T$'s instructions, $L$ then attemps to train a policy $\pi^L$ whose feature expectations approximate those of $\pi^T$. Note that, if this is successful, the performance of $\pi^L$ is close to that of $\pi^T$ due to the form of the reward function.

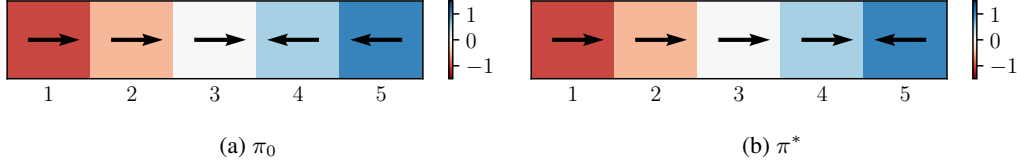

(a) $\pi_0$                               (b) $\pi^*$

Figure 1: A simple example to illustrate the challenges arising when teaching under worldview mismatch. We consider an MDP in which $S = \{s_1, \ldots, s_5\}$ is the set of cells in the gridworld displayed, $A = \{\leftarrow, \rightarrow\}$, and $R(s) = \langle \mathbf{w}^*, \phi(s) \rangle$ with feature map $\phi : S \to \mathbb{R}^5$ taking $s_i$ to the one-hot vector $e_i \in \mathbb{R}^5$. The initial state distribution is uniform and the transition dynamics are deterministic. More specifically, when the agent takes action $\rightarrow$ (resp. $\leftarrow$), it moves to the neighboring cell to the right (resp. left); when the agent is in the rightmost (resp. leftmost) cell, the action $\rightarrow$ (resp. $\leftarrow$) is not permitted. The reward weights are given by $\mathbf{w}^* = (-1, -0.5, 0, 0.5, 1)^T \in \mathbb{R}^5$ up to normalization; the values $R(s_i) = \langle \mathbf{w}^*, \phi(s_i) \rangle$ are also encoded by the colors of the cells. The policy $\pi^*$ in (b) is the optimal policy with respect to the true reward function. Assuming that the learner $L$ only observes the feature corresponding to the central cell, i.e., $A^L = (0 \quad 0 \quad 1 \quad 0 \quad 0) \in \mathbb{R}^{1 \times 5}$, the policy $\pi_0$ in (a) is a better teaching policy in the worst-case sense. See the main text for a detailed description.

We assume that $L$ has access to an algorithm that enables her to do the following: Whenever she is given sufficiently many demonstrations sampled from a policy $\pi^T$, she is able to find a policy $\pi^L$ whose feature expectations in *her* worldview approximate those of $\pi^T$, i.e., $A^L \mu(\pi^L) \approx A^L \mu(\pi^T)$. Examples for algorithms that $L$ could use to that end are the algorithms in [Abbeel and Ng, 2004] and [Ziebart et al., 2008] which are based on IRL. The following discussion does not require any further specification of what precise algorithm $L$ uses in order to match feature expectations.

**Challenges when teaching under worldview mismatch.** If there was no mismatch in the world-view (i.e., if $A^L$ was the identity matrix in $\mathbb{R}^{k \times k}$), then the teacher could simply provide demonstrations from the optimal policy $\pi^*$ to achieve the desired objective. However, the example in Figure 1 illustrates that this is not the case when there is a mismatch between the worldviews.

For the MDP in Figure 1, assume that the teacher provides demonstrations using $\pi^T = \pi^*$, which moves to the rightmost cell as quickly as possible and then alternates between cells 4 and 5 (see Figure 1b). Note that the policy $\widetilde{\pi}^*$ which moves to the leftmost cell as quickly as possible and then alternates between cells 1 and 2, has the same feature expectations as $\pi^*$ in the learner's worldview; in fact, $\widetilde{\pi}^*$ is the unique policy other than $\pi^*$ with that property (provided we restrict to deterministic policies). As the teacher is unaware of the internal workings of the learner, she has no control over which of these two policies the learner will eventually learn by matching feature expectations.

However, the teacher can ensure that the learner achieves better performance in a worst case sense by providing demonstrations tailored to the problem at hand. In particular, assume that the teacher uses $\pi^T = \pi_0$, the policy shown in Figure 1a, which moves to the central cell as quickly as possible and then alternates between cells 3 and 4. The only other policy $\widetilde{\pi}_0$ with which the learner could match the feature expectations of $\pi_0$ in her worldview (restricting again to deterministic policies) is the one that moves to the central cell as quickly as possible and then alternates between states 2 and 3.

Note that $R(\pi^*) > R(\pi_0) > R(\widetilde{\pi}_0) > R(\widetilde{\pi}^*)$, and hence $\pi_0$ is a better teaching policy than $\pi^*$ regarding the performance that a learner matching feature expectations in her worldview achieves in the worst case. In particular, this example shows that providing demonstrations from the truly optimal policy $\pi^*$ does not guarantee that the learner's policy achieves good performance in general.

## 4 Teaching Risk

**Definition of teaching risk.** The fundamental problem in the setting described in Section 3 is that two policies $\pi_0, \pi_1$ that perform equally well with respect to any estimate $s \mapsto \langle \mathbf{w}^L, A^L \phi(s) \rangle$ that $L$ may have of the reward function, may perform very differently with respect to the true reward function. Hence, even if $L$ is able to imitate the behaviour of the teacher well in *her* worldview, there

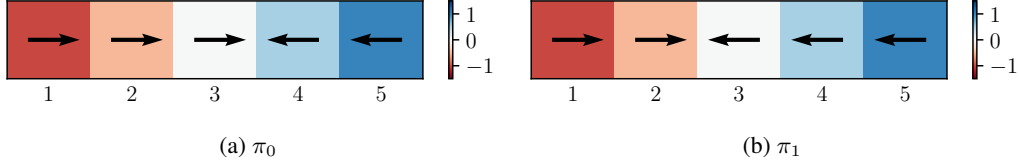

(a) $\pi_0$  (b) $\pi_1$

Figure 2: Two policies in the environment introduced in Figure 1 (the policy $\pi_0$ here is identical to the one in Figure 1(a)). We assume again that $L$ can only observe the feature corresponding to the central cell. Provided that the initial state distribution is uniform, the feature expectations of $\pi_0$ and $\pi_1$ in $L$'s worldview are equal, and hence these policies perform equally well with respect to any estimate of the reward function that $L$ may have. In fact, both look optimal to $L$ if she assumes that the central cell carries positive reward. However, their performance with respect to the true reward function is positive for $\pi_0$ but negative for $\pi_1$. This illustrates that, if all we know about $L$ is that she matches feature counts in her worldview, we can generally not give good performance guarantees for the policy she finds.

is genenerally no guarantee on how good her performance is with respect to the true reward function. For an illustration, see Figure 2.

To address this problem, we define the following quantity: The **teaching risk** for a given worldview $A^L$ with respect to reward weights $\mathbf{w}^*$ is

$$\rho(A^L; \mathbf{w}^*) := \max_{v \in \ker A^L, \|v\| \leq 1} \langle \mathbf{w}^*, v \rangle. \tag{1}$$

Here $\ker A^L = \{v \in \mathbb{R}^k \mid A^L v = 0\}$ and $\ker \mathbf{w}^* = \{v \in \mathbb{R}^k \mid \langle \mathbf{w}^*, v \rangle = 0\}$ denote the kernels of $A^L$ resp. $\langle \mathbf{w}^*, \cdot \rangle$. Geometrically, $\rho(A^L; \mathbf{w}^*)$ is the cosine of the angle between $\ker A^L$ and $\mathbf{w}^*$; in other words, $\rho(A^L; \mathbf{w}^*)$ measures the degree to which $\ker A^L$ deviates from satisfying $\ker A^L \subseteq \ker \mathbf{w}^*$. Yet another way of characterizing the teaching risk is as $\rho(A^L; \mathbf{w}^*) = \|\operatorname{pr}(\mathbf{w}^*)\|$, where $\operatorname{pr} : \mathbb{R}^k \to \ker A^L$ denotes the orthogonal projection onto $\ker A^L$.

**Significance of teaching risk.** To understand the significance of the teaching risk in our context, assume that $L$ is able to find a policy $\pi^L$ which matches the feature expectations of $T$'s (not necessarily optimal) policy $\pi^T$ perfectly in her worldview, which is equivalent to $\mu(\pi^T) - \mu(\pi^L) \in \ker A^L$. Directly from the definition of the teaching risk, we see that the gap between their performances with respect to the *true* reward function satisfies

$$|\langle \mathbf{w}^*, \mu(\pi^T) - \mu(\pi^L) \rangle| \leq \rho(A^L; \mathbf{w}^*) \cdot \|\mu(\pi^T) - \mu(\pi^L)\|, \tag{2}$$

with equality if $\mu(\pi^T) - \mu(\pi^L)$ is proportional to a vector $v$ realizing the maximum in (1). If the teaching risk is large, this performance gap can generally be large as well. This motivates the interpretation of $\rho(A^L; \mathbf{w}^*)$ as a measure of the risk when teaching the task modelled by an MDP with reward weights $\mathbf{w}^*$ to a learner whose worldview is represented by $A^L$.

On the other hand, smallness of the teaching risk implies that this performance gap cannot be too large. The following theorem, proven in the extended version of this paper [Haug et al., 2018], generalizes the bound in (2) to the situation in which $\pi^L$ only approximates the feature expectations of $\pi^T$.

**Theorem 1.** *Assume that* $\|A^L(\mu(\pi^T) - \mu(\pi^L))\| < \varepsilon$. *Then the gap between the true performances of* $\pi^L$ *and* $\pi^T$ *satisfies*

$$|\langle \mathbf{w}^*, \mu(\pi^T) - \mu(\pi^L) \rangle| < \frac{\varepsilon}{\sigma(A^L)} + \rho(A^L; \mathbf{w}^*) \cdot \operatorname{diam} \mu(\Pi)$$

*with* $\sigma(A^L) = \min_{v \perp \ker A^L, \|v\| = 1} \|A^L v\|$.

Theorem 1 shows the following: If $L$ imitates $T$'s behaviour well in her worldview (meaning that $\varepsilon$ can be chosen small) and if the teaching risk $\rho(A^L; \mathbf{w}^*)$ is sufficiently small, then $L$ will perform nearly as well as $T$ with respect to the true reward. In particular, if $T$'s policy is optimal, $\pi^T = \pi^*$, then $L$'s policy $\pi^L$ is guaranteed to be near-optimal.

**Algorithm 1** TRGREEDY: Feature- and demo-based teaching with TR-greedy feature selection

---

**Require:** Reward vector $\mathbf{w}^*$, set of teachable features $\mathcal{F}$, feature budget $B$, initial worldview $A^L$,
  teacher policy $\pi^T$, initial learner policy $\pi^L$, performance threshold $\varepsilon$.
  **for** $i = 1, \ldots, B$ **do**
    **if** $|\langle \mathbf{w}^*, \mu(\pi^L) \rangle - \langle \mathbf{w}^*, \mu(\pi^T) \rangle| > \varepsilon$ **then**
      $f \leftarrow \arg\min_{f \in \mathcal{F}} \rho(A^L \oplus \langle f, \cdot \rangle; \mathbf{w}^*)$       ▷ $T$ selects feature to teach
      $A^L \leftarrow A^L \oplus \langle f, \cdot \rangle$                ▷ $L$'s worldview gets updated
      $\pi^L \leftarrow \text{LEARNING}(\pi^L, A^L \mu(\pi^T))$              ▷ $L$ trains a new policy
    **else**
      **return** $\pi^L$
    **end if**
  **end for**
  **return** $\pi^L$

---

The quantity $\sigma(A^L)$ appearing in Theorem 1 is a bound on the amount to which $A^L$ distorts lengths of vectors in the orthogonal complement of $\ker A^L$. Note that $\sigma(A^L)$ is independent of the teaching risk, in the sense that one can change it, e.g., by rescaling $A^L \to \alpha A^L$ by some $\alpha \in \mathbb{R}_+$, without changing the teaching risk.

**Teaching risk as obstruction to recognizing optimality.**  We now provide a second motivation for the consideration of the teaching risk, by interpreting it as a quantity that measures the degree to which truly optimal policies deviate from looking optimal to $L$. We make the technical assumption that $\mu(\Pi)$ is the closure of a bounded open set with smooth boundary $\partial \mu(\Pi)$ (this will only be needed for the proofs). Our first observation is the following:

**Proposition 1.** *Let $\pi^*$ be a policy which is optimal for $s \mapsto \langle \mathbf{w}^*, \boldsymbol{\phi}(s) \rangle$. If $\rho(A^L; \mathbf{w}^*) > 0$, then $\pi^*$ is suboptimal with respect to* any *choice of reward function $s \mapsto \langle \mathbf{w}, A^L \boldsymbol{\phi}(s) \rangle$ with $\mathbf{w} \in \mathbb{R}^\ell$.*

In view of Proposition 1, a natural question is whether we can bound the suboptimality, in $L$'s view, of a truly optimal policy in terms of the teaching risk. The following theorem provides such a bound:

**Theorem 2.** *Let $\pi^*$ be a policy which is optimal for $s \mapsto \langle \mathbf{w}^*, \boldsymbol{\phi}(s) \rangle$. There exists a unit vector $\mathbf{w}_L^* \in \mathbb{R}^\ell$ such that*

$$\max_{\mu \in \mu(\Pi)} \left\langle \mathbf{w}_L^*, A^L \mu \right\rangle - \left\langle \mathbf{w}_L^*, A^L \mu(\pi^*) \right\rangle \leq \frac{\operatorname{diam} \mu(\Pi) \cdot \|A^L\| \cdot \rho(A^L; \mathbf{w}^*)}{\sqrt{1 - \rho(A^L; \mathbf{w}^*)^2}},$$

*where $\|A^L\| = \max_{v \in \mathbb{R}^k, \|v\|=1} \|A^L v\|$.*

Proofs of Proposition 1 and Theorem 2 are given in the extended version of this paper [Haug et al., 2018]. Note that the expression on the right hand side of the inequality in Theorem 2 tends to 0 as $\rho(A^L; \mathbf{w}^*) \to 0$, provided $\|A^L\|$ is bounded. Theorem 2 therefore implies that, if $\rho(A^L; \mathbf{w}^*)$ is small, a truly optimal policy $\pi^*$ is near-optimal for *some* choice of reward function linear in the features $L$ observes, namely, the reward function $s \mapsto \langle \mathbf{w}_L^*, \boldsymbol{\phi}(s) \rangle$ with $\mathbf{w}_L^* \in \mathbb{R}^\ell$ the vector whose existence is claimed by the theorem.

## 5  Teaching

**Feature teaching.**  The discussion in the last section shows that, under our assumptions on how $L$ learns, a teaching scheme in which $T$ solely provides demonstrations to $L$ can generally, i.e., without any assumption on the teaching risk, not lead to reasonable guarantees on the learner's performance with respect to the true reward. A natural strategy is to introduce additional teaching operations by which the teacher can update $L$'s worldview $A^L$ and thereby decrease the teaching risk.

The simplest way by which the teacher $T$ can change $L$'s worldview is by informing her about features $f \in \mathbb{R}^k$ that are relevant to performing well in the task, thus causing her to update her worldview $A^L \colon \mathbb{R}^k \to \mathbb{R}^\ell$ to

$$A^L \oplus \langle f, \cdot \rangle \colon \mathbb{R}^k \to \mathbb{R}^{\ell+1}.$$

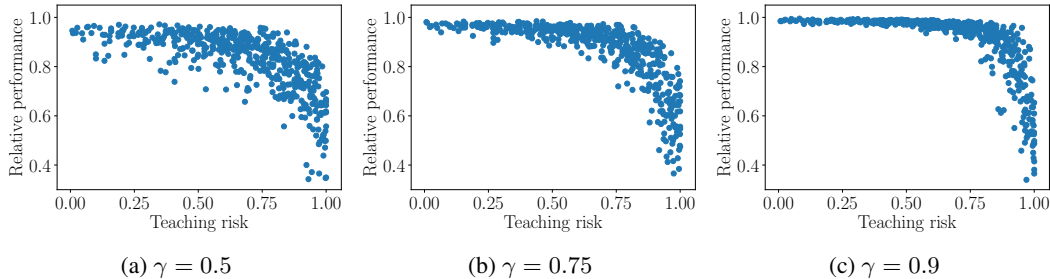

(a) $\gamma = 0.5$          (b) $\gamma = 0.75$          (c) $\gamma = 0.9$

Figure 3: Performance vs. teaching risk. Each point in the plots shows the relative performance that a learner $L$ with a random worldview matrix $A^L$ achieved after one round of learning and the teaching risk of $A^L$. For all plots, a gridworld with $N = 20$, $n = 2$ was used. The reward vector $\mathbf{w}^*$ was sampled randomly in each round. (a)–(c) correspond to different values of the discount factor $\gamma$.

Viewing $A^L$ as a matrix, this operation appends $f$ as a row to $A^L$. (Strictly speaking, the feature that is thus provided is $s \mapsto \langle f, \phi(s) \rangle$; we identify this map with the vector $f$ in the following and thus keep calling $f$ a "feature".)

This operation has simple interpretations in the settings we are interested in: If $L$ is a human learner, "teaching a feature" could mean making $L$ aware that a certain quantity, which she might not have taken into account so far, is crucial to achieving high performance. If $L$ is a machine, such as an autonomous car or a robot, it could mean installing an additional sensor.

**Teachable features.**   Note that if $T$ could provide arbitrary vectors $f \in \mathbb{R}^k$ as new features, she could always, no matter what $A^L$ is, decrease the teaching risk to zero in a single teaching step by choosing $f = \mathbf{w}^*$, which amounts to telling $L$ the true reward function. We assume that this is not possible, and that instead only the elements of a fixed finite set of *teachable features*

$$\mathcal{F} = \{f_i \mid i \in I\} \subset \mathbb{R}^k$$

can be taught. In real-world applications, such constraints could come from the limited availability of sensors and their costs; in the case that $L$ is a human, they could reflect the requirement that features need to be interpretable, i.e., that they can only be simple combinations of basic observable quantities.

**Greedy minimization of teaching risk.**   Our basic teaching algorithm TRGREEDY (Algorithm 1) works as follows: $T$ and $L$ interact in rounds, in each of which $T$ provides $L$ with the feature $f \in \mathcal{F}$ which reduces the teaching risk of $L$'s worldview with respect to $\mathbf{w}^*$ by the largest amount. $L$ then trains a policy $\pi^L$ with the goal of imitating her current view $A^L \mu(\pi^T)$ of the feature expectations of the teacher's policy; the LEARNING algorithm she uses could be the apprenticeship learning algorithm from [Abbeel and Ng, 2004].

**Computation of the teaching risk.**   The computation of the teaching risk required of $T$ in every round of Algorithm 1 can be performed as follows: One first computes the orthogonal complement of $\ker A^L \cap \ker \mathbf{w}^*$ in $\mathbb{R}^k$ and intersects that with $\ker A^L$, thus obtaining (generically) a 1-dimensional subspace $\lambda = (\ker A^L \cap \ker \mathbf{w}^*)^{\perp} \cap \ker A^L$ of $\mathbb{R}^k$; this can be done using SVD. The teaching risk is then $\rho(A^L; \mathbf{w}^*) = \langle \mathbf{w}^*, v^{\perp} \rangle$ with $v^{\perp}$ the unique unit vector in $\lambda$ with $\langle \mathbf{w}^*, v^{\perp} \rangle > 0$.

## 6   Experiments

Our experimental setup is similar to the one in [Abbeel and Ng, 2004], i.e., we use $N \times N$ gridworlds in which non-overlapping square regions of neighbouring cells are grouped together to form $n \times n$ macrocells for some $n$ dividing $N$. The state set $S$ is the set of gridpoints, the action set is $A = \{\leftarrow, \rightarrow, \uparrow, \downarrow\}$, and the feature map $\phi \colon S \to \mathbb{R}^k$ maps a gridpoint belonging to macrocell $i \in \{1, \ldots, (N/n)^2\}$ to the one-hot vector $e_i \in \mathbb{R}^k$; the dimension of the "true" feature space is therefore $k = (N/n)^2$. Note that these gridworlds satisfy the quite special property that for states $s \neq s'$, we either have $\phi(s) = \phi(s')$ (if $s, s'$ belong to the same macrocell), or $\phi(s) \perp \phi(s')$. The reward weights $\mathbf{w}^* \in \mathbb{R}^k$ are sampled randomly for all experiments unless mentioned otherwise. As

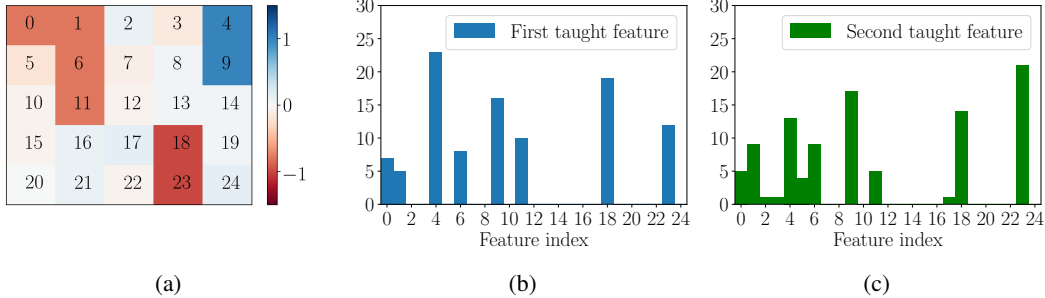

(a)    (b)    (c)

Figure 4: Gridworld with $N = 10, n = 2$. The colors in (a) indicate the reward of the corresponding macrocell, with blue meaning positive and red meaning negative reward. The numbers within each macrocell correspond to the feature index. The histograms in (b) and (c) show how often, in a series of 100 experiments, each feature was selected as the first resp. second feature to be taught to a learner with a random 5-dimensional initial worldview.

the LEARNING algorithm within Algorithm 1, we use the *projection version* of the apprenticeship learning algorithm from [Abbeel and Ng, 2004].

**Performance vs. teaching risk.**    The plots in Figure 3 illustrate the significance of the teaching risk for the problem of teaching a learner under worldview mismatch. To obtain these plots, we used a gridworld with $N = 20$, $n = 2$; for each value $\ell \in [1, 100]$, we sampled five random worldview matrices $A^L \in \mathbb{R}^{\ell \times 100}$, and let $L$ train a policy $\pi^L$ using the projection algorithm in [Abbeel and Ng, 2004], with the goal of matching the feature expectations $\mu(\pi^T)$ corresponding to an optimal policy $\pi^T$ for a reward vector $\mathbf{w}^*$ that was sampled randomly in each round. Each point in the plots corresponds to one such experiment and shows the relative performance of $\pi_L$ after the training round vs. the teaching risk of $L$'s worldview matrix $A^L$.

All plots in Figure 3 show that the variance of the learner's performance decreases as the teaching risk decreases. This supports our interpretation of the teaching risk as a measure of the potential gap between the performances of $\pi^L$ and $\pi^T$ when $L$ matches the feature expectations of $\pi^T$ in her worldview. The plots also show that the bound for this gap provided in Theorem 1 is overly conservative in general, given that $L$'s performance is often high and has small variance even if the teaching risk is relatively large.

The plots indicate that for larger $\gamma$ (i.e., less discounting), it is easier for $L$ to achieve high performance even if the teaching risk is large. This makes intuitive sense: If there is a lot of discounting, it is important to reach high reward states quickly in order to perform well, which necessitates being able to recognize where these states are located, which in turn requires the teaching risk to be small. If there is little discounting, it is sufficient to know the location of *some* maybe distant reward state, and hence even a learner with a very deficient worldview (i.e., high teaching risk) can do well in that case.

**Small gridworlds with high reward states and obstacles.**    We tested TRGREEDY (Algorithm 1) on gridworlds such as the one in Figure 4a, with a small number of states with high positive rewards, some obstacle states with high negative rewards, and all other states having rewards close to zero. The histograms in Figures 4b and 4c show how often each of the features was selected by the algorithm as the first resp. second feature to be taught to the learner in 100 experiments, in each of which the learner was initialized with a random 5-dimensional worldview. In most cases, the algorithm first selected the features corresponding to one of the high reward cells 4 and 9 or to one of the obstacle cells 18 and 23, which are clearly those that the learner must be most aware of in order to achieve high performance.

**Comparison of algorithms.**    We compared the performance of TRGREEDY (Algorithm 1) to two variants of the algorithm which are different only in how the features to be taught in each round are selected: The first variant, RANDOM, simply selects a random feature $f \in \mathcal{F}$ from the set of all teachable features. The second variant, PERFGREEDY, greedily selects the feature that will lead to the best performance in the next round among all $f \in \mathcal{F}$ (computed by simulating the teaching process for each feature and evaluating the corresponding learner).

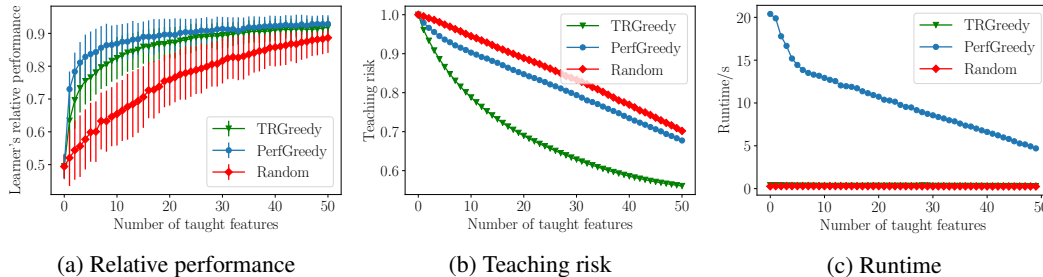

(a) Relative performance      (b) Teaching risk      (c) Runtime

Figure 5: Comparison of TRGREEDY vs. PERFGREEDY vs. RANDOM. The plots show (a) the relative performance that the learner achieved after each round of feature teaching and training a policy, (b) the teaching risk after each such step, and (c) the runtime required to perform each step. We averaged over 100 experiments, in each of which a new random gridworld of size $(N, n) = (20, 2)$ and a new set $\mathcal{F}$ of randomly selected features with $|\mathcal{F}| = 70$ were sampled; the bars in the relative performance plot indicate the standard deviations. The discount factor used was $\gamma = 0.9$ in all cases.

The plots in Figure 5 show, for each of the three algorithms, the relative performance with respect to the true reward function $s \mapsto \langle \mathbf{w}^*, \boldsymbol{\phi}(s) \rangle$ that the learner achieved after each round of feature teaching and training a policy $\pi^L$, as well as the corresponding teaching risks and runtimes, plotted over the number of features taught. The relative performance of the learner's policy $\pi^L$ was computed as $(R(\pi^L) - \min_\pi R(\pi))/(\max_\pi R(\pi) - \min_\pi R(\pi))$.

We observed in all our experiments that TRGREEDY performed significantly better than RANDOM. While the comparison between TRGREEDY and PERFGREEDY was slightly in favour of the latter, one should note that a teacher $T$ running PERFGREEDY must simulate a learning round of $L$ for all features $f \in \mathcal{F}$ not yet taught, which presupposes that $T$ knows $L$'s learning algorithm, and which also leads to very high runtime. If $T$ only knows that $L$ is able to match (her view of) the feature expectations of $T$'s demonstrations and $T$ simulates $L$ using *some* algorithm capable of this, there is no guarantee that $L$ will perform as well as $T$'s simulated counterpart, as there may be a large discrepancy between the true performances of two policies which in $L$'s view have the same feature expectations. In contrast, TRGREEDY relies on much less information, namely the kernel of $A^L$, and in particular is agnostic to the precise learning algorithm that $L$ uses to approximate feature counts.

## 7 Conclusions and Outlook

We presented an approach to dealing with the problem of *worldview mismatch* in situations in which a learner attempts to find a policy matching the feature counts of a teacher's demonstrations. We introduced the *teaching risk*, a quantity that depends on the worldview of the learner and the true reward function and which (1) measures the degree to which policies which are optimal from the point of view of the learner can be suboptimal from the point of view of the teacher, and (2) is an obstruction for truly optimal policies to look optimal to the learner. We showed that under the condition that the teaching risk is small, a learner matching feature counts using, e.g., standard IRL-based methods is guaranteed to learn a near-optimal policy from demonstrations of the teacher even under worldview mismatch.

Based on these findings, we presented our teaching algorithm TRGREEDY, in which the teacher updates the learner's worldview by teaching her features which are relevant for the true reward function in a way that greedily minimizes the teaching risk, and then provides her with demonstrations based on which she learns a policy using any suitable algorithm. We tested our algorithm in gridworld settings and compared it to other ways of selecting features to be taught. Experimentally, we found that TRGREEDY performed comparably to a variant which selected features based on greedily maximizing performance, and consistently better than a variant with randomly selected features.

We plan to investigate extensions of our ideas to nonlinear settings and to test them in more complex environments in future work. We hope that, ultimately, such extensions will be applicable in real-world scenarios, for example in systems in which human expert knowledge is represented as a reward function, and where the goal is to teach this expert knowledge to human learners.

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
