[Supplementary Material · supplementary.pdf]

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

$ the orthogonal projection onto $\ker A^L$ and let $v = \mathrm{pr}(\mu(\pi^L) - \mu(\pi^T))$. Note that we have $v + \mu(\pi^T) - \mu(\pi^L) \in (\ker A^L)^\perp$ and $\|v\| \leq \|\mu(\pi^T) - \mu(\pi^L)\| \leq \mathrm{diam}\, \mu(\Pi)$. It follows that

$$\|v + \mu(\pi^T) - \mu(\pi^L)\| \leq \frac{1}{\sigma(A^L)} \|A^L(v + \mu(\pi^T) - \mu(\pi^L))\|$$

$$= \frac{1}{\sigma(A^L)} \|A^L(\mu(\pi^T) - \mu(\pi^L))\|$$

$$< \frac{\varepsilon}{\sigma(A^L)},$$

using the definition of $\sigma(A^L)$, the fact that $v \in \ker A^L$, and the assumption that $\|A^L(\mu(\pi^T) - \mu(\pi^L))\| < \varepsilon$. We then obtain

$$|\langle \mathbf{w}^*, \mu(\pi^T)\rangle - \langle \mathbf{w}^*, \mu(\pi^L)\rangle| \leq |\langle \mathbf{w}^*, v + \mu(\pi^T) - \mu(\pi^L)\rangle| + |\langle \mathbf{w}^*, v\rangle|$$

$$\leq \|v + \mu(\pi^T) - \mu(\pi^L)\| + \|v\| \cdot \rho(A^L; \mathbf{w}^*)$$

$$\leq \frac{\varepsilon}{\sigma(A^L)} + \mathrm{diam}\, \mu(\Pi) \cdot \rho(A^L; \mathbf{w}^*)$$

using the triangle inequality, the Cauchy-Schwarz inequality and the definition of $\rho(A^L; \mathbf{w}^*)$, and the estimates above. □

# B  Proof of Proposition 1

Figure 6: A situation in which $\rho(A^L; \mathbf{w}^*) > 0$: Here, $A^L$ is the projection on the horizontal axis. The points in $\partial\mu(\Pi)$ which $A^L$ maps to the boundary of $A^L\mu(\Pi)$, and which therefore appear optimal to $L$, are the two points marked by ◆, at which the normal vector to $\mu(\Pi)$ is contained in $(\ker A^L)^\perp$; these are precisely the points which are optimal for some $\mathbf{w}^*$ with $\rho(A^L; \mathbf{w}^*) = 0$ (namely, $\mathbf{w}^* = (\pm 1, 0)$). All other points in $\partial\mu(\Pi)$ get mapped by $A^L$ to the interior of $A^L\mu(\Pi)$ and therefore appear suboptimal for any choice of reward function $L$ might consider.

*Proof of Proposition 1.* As mentioned in the main text, we assume that the set $\mu(\Pi) \subset \mathbb{R}^k$ is the closure of a bounded open set and has a smooth boundary $\partial\mu(\Pi)$.

Finding the feature expectations $\mu(\pi^*)$ of a policy $\pi^*$ which is optimal with respect to $s \mapsto \langle \mathbf{w}^*, \phi(s)\rangle$ is equivalent to maximizing the linear map $\mu \mapsto \langle \mathbf{w}^*, \mu\rangle$ over $\mu(\Pi) \subset \mathbb{R}^k$, whose gradient is $\mathbf{w}^* \neq 0$. It follows that these feature expectations need to satisfy the following conditions:

1. $\mu(\pi^*)$ lies on the boundary $\partial\mu(\Pi)$,

2. $\mathbf{w}^*$ is normal to $\partial\mu(\Pi)$ at $\mu(\pi^*)$.

The second condition is equivalent to saying that the tangent space to $\mu(\Pi)$ at $\mu(\pi^*)$ is $\ker \mathbf{w}^*$.

Assume now that $\rho(A^L; \mathbf{w}^*) > 0$. This is equivalent to saying that $\ker A^L \not\subseteq \ker \mathbf{w}^*$, i.e., to saying that $\ker A^L$ is not tangent to $\partial\mu(\Pi)$ at $\mu(\pi^*)$. That implies that there exist some $v \in \ker A^L$ such that $\mu(\pi^*) + v$ is contained in the interior of $\mu(\Pi)$, which means that a sufficiently small ball around $\mu(\pi^*) + v$ is contained in $\mu(\Pi)$. In particular, a small ball around $\mu(\pi^*) + v$ in the affine space $\mu(\pi^*) + v + (\ker A_L)^\perp$ is entirely contained in $\mu(\Pi)$. This implies that $A^L(\mu(\pi^*)) = A^L(\mu(\pi^* + v))$ is contained in the interior of $A^L\mu(\Pi)$, i.e., not in the boundary $\partial A^L(\mu(\pi^*))$. Therefore $\pi^*$ is suboptimal with respect to any choice of reward function $s \mapsto \langle \mathbf{w}, A^L\phi(s) \rangle$ with $\mathbf{w} \in \mathbb{R}^\ell$. $\qquad\square$

## C   Proof of Theorem 2

*Proof of Theorem 2.* The assumption that $\pi^*$ is optimal for the reward function $s \mapsto \langle \mathbf{w}^*, \phi(s) \rangle$ implies that $\langle \mathbf{w}^*, \mu - \mu(\pi^*) \rangle \leq 0$ for all $\mu \in \mu(\Pi)$. By decomposing $\mathbf{w}^*$ as $\mathbf{w}^* = \mathrm{pr}(\mathbf{w}^*) + \mathrm{pr}^\perp(\mathbf{w}^*)$, where $\mathrm{pr} : \mathbb{R}^k \to \ker A^L$ denotes the orthogonal projection onto $\ker A^L$ and $\mathrm{pr}^\perp : \mathbb{R}^k \to (\ker A^L)^\perp$ the orthogonal projection onto $(\ker A^L)^\perp$, we obtain

$$\langle \mathrm{pr}(\mathbf{w}^*), \mu - \mu(\pi^*) \rangle + \langle \mathrm{pr}^\perp(\mathbf{w}^*), \mu - \mu(\pi^*) \rangle \leq 0. \tag{3}$$

The first summand can be bounded as follows:

$$\begin{aligned}
\langle \mathrm{pr}(\mathbf{w}^*), \mu - \mu(\pi^*) \rangle &\geq -\|\mu - \mu(\pi^*)\| \cdot \|\mathrm{pr}(\mathbf{w}^*)\| \\
&= -\|\mu - \mu(\pi^*)\| \cdot \rho(A^L; \mathbf{w}^*) \\
&\geq -\operatorname{diam}\mu(\Pi) \cdot \rho(A^L; \mathbf{w}^*),
\end{aligned} \tag{4}$$

using the Cauchy-Schwarz inequality and the fact that $\|\mathrm{pr}(\mathbf{w}^*)\| = \rho(A^L; \mathbf{w}^*)$. By combining estimates (3) and (4), we obtain

$$\langle \mathrm{pr}^\perp(\mathbf{w}^*), \mu - \mu(\pi^*) \rangle \leq \operatorname{diam}\mu(\Pi) \cdot \rho(A^L; \mathbf{w}^*). \tag{5}$$

Denote now by $(A^L)^+$ the Moore-Penrose pseudoinverse of $A^L$, and by $X := ((A^L)^+)^T$ its transpose. We have

$$\begin{aligned}
\langle \mathrm{pr}^\perp(\mathbf{w}^*), \mu - \mu(\pi^*) \rangle &= \langle \mathbf{w}^*, \mathrm{pr}^\perp(\mu - \mu(\pi^*)) \rangle \\
&= \langle \mathbf{w}^*, (A^L)^+ A^L \mathrm{pr}^\perp(\mu - \mu(\pi^*)) \rangle \\
&= \langle X\mathbf{w}^*, A^L \mathrm{pr}^\perp(\mu - \mu(\pi^*)) \rangle \\
&= \langle X\mathbf{w}^*, A^L(\mu - \mu(\pi^*)) \rangle,
\end{aligned} \tag{6}$$

where the second equality uses the fact that the restriction of $(A^L)^+ A^L$ to $(\ker A^L)^\perp$ is the identity (in fact, $(A^L)^+ A^L = \mathrm{pr}^\perp$, a general property of Moore-Penrose pseudoinverses). Setting $\mathbf{w}_L^* := \frac{1}{\|X\mathbf{w}^*\|} X\mathbf{w}^*$ and combining inequality (5) with (6), we obtain

$$\langle \mathbf{w}_L^*, A^L(\mu - \mu(\pi^*)) \rangle \leq \frac{\operatorname{diam}\mu(\Pi) \cdot \rho(A^L; \mathbf{w}^*)}{\|X\mathbf{w}^*\|}. \tag{7}$$

We now estimate the term $\|X\mathbf{w}^*\|$:

$$\begin{aligned}
\|X\mathbf{w}^*\| &= \max_{v \in \mathbb{R}^\ell, \|v\|=1} \langle X\mathbf{w}^*, v \rangle \\
&= \max_{v \in \mathbb{R}^\ell, \|v\|=1} \langle \mathbf{w}^*, (A^L)^+ v \rangle \\
&\geq \frac{\langle \mathbf{w}^*, (A^L)^+ A^L \mathrm{pr}^\perp(\mathbf{w}^*) \rangle}{\|A^L \mathrm{pr}^\perp(\mathbf{w}^*)\|} \\
&= \frac{\|\mathrm{pr}^\perp(\mathbf{w}^*)\|^2}{\|A^L \mathrm{pr}^\perp(\mathbf{w}^*)\|} \\
&= \frac{\|\mathrm{pr}^\perp(\mathbf{w}^*)\|}{\|A^L \frac{\mathrm{pr}^\perp(\mathbf{w}^*)}{\|\mathrm{pr}^\perp(\mathbf{w}^*)\|}\|} \\
&\geq \frac{\|\mathrm{pr}^\perp(\mathbf{w}^*)\|}{\|A^L\|}.
\end{aligned} \tag{8}$$

Since $\|\operatorname{pr}^{\perp}(\mathbf{w}^*)\| = \sqrt{\|\mathbf{w}^*\|^2 - \|\operatorname{pr}(\mathbf{w}^*)\|^2} = \sqrt{1 - \rho(A^L; \mathbf{w}^*)^2}$, combining (7) and (8) yields

$$\langle \mathbf{w}_L^*, A^L(\mu - \mu(\pi^*)) \rangle \leq \frac{\operatorname{diam} \mu(\Pi) \cdot \|A^L\| \cdot \rho(A^L; \mathbf{w}^*)}{\sqrt{1 - \rho(A^L; \mathbf{w}^*)^2}}$$

This holds for all $\mu \in \mu(\Pi)$, and hence we can maximize over $\mu$ to obtain the statement claimed in Theorem 2. $\qquad\square$