[Reviews · NeurIPS 2018]

Reviewer 1



After response: The authors' response helped clarify the computability of the risk metric for me. More experiments and discussion would improve the paper but after reading the response and other reviews I am raising my score. ***************************** This paper introduces the problem of feature mismatch between a learner and teacher in the inverse RL problem (IRL). The problem is that if the learner and teacher represent the reward with different features than the learner may not be able to learn the teacher's reward function and thus perform suboptimally if it attempts to learn by matching feature counts. The paper quantifies this problem by defining the "teaching risk" and demonstrates that minimizing this quantity can improve learner performance. An algorithm is introduced that uses this quantity to select new features for the learner to use and it is shown to empirically lead to the learner being able to learn with less features than other baselines. This paper presents an interesting problem that is likely to happen in real world applications of IRL. Furthermore, quantifying this problem in a way that corresponds to performance loss is interesting and is shown to be useful (though I have some questions about computability). To the best of my knowledge, this paper is the first to study this problem in the IRL setting and opens up an interesting direction for future research. My main concern with the paper is that it lacks clarity in some places (see below) and has little discussion of results or future work. Overall, I lean towards not accepting this paper but think it could be a stronger submission with appropriate revision. My main technical concern is the computability of the teaching risk. Is the teaching risk computable in practice or is it just theoretically interesting? IRL is usually motivated by situations where w* is unknown but computing \rho seems to require w*. If w* is unknown, how can we compute the teaching risk in practice? In terms of clarity, the paper relies on figures to make several of its key points. However, the description of the figures is only contained in the main text. The figures should have a complete description of their content (preferably in the caption). Right now the description is interspersed with remarks on what they demonstrate. It should be clear what is being shown before the reader is asked to accept conclusions about what they show. In my opinion, figures 1, 2, and 3 don't illustrate much. It would be helpful for Figure 3 if the figure was described instead of simply saying, "Figure 3 shows why proposition 1 is true." Where possible, it would help to see intuition before equations. For example, in "Basic Teaching Strategy" (lines 123-128) it would be useful to have the description of (1) before (1). The paper lacks any discussion of future work and has only limited discussion of the empirical results. Since the paper is proposing a new problem, it would be good to have the paper also outline what the next steps are in addressing the problem. Minor comments: 18: an reinforcement -> a reinforcement 62: recently been recently 67: have -> has if using citation number instead of names. Citations: Using numbers in place of names means the reader has to constantly turn to the references section. Saying "Name et al." is easier on the reader. 172: is vanishes -> vanishes 179 (and elsewhere): garantee -> guarantee 187 (and elsewhere): the use of "resp." is sometimes hard to parse what is being referred to. 283: better a -> better than a 273: optiaml -> optimal Supplementary material: It would be useful to have some section headers bolded. Line 140: policies pi_0, pi_1 have not been introduced at this point. Line 148: Is something missing in the definition of ker A^L

Reviewer 2



1. Summary - This paper introduces a new IRL scenario where the learner has a different “world-view” (features) compared to the teacher. A key observation is that the optimal policy with respect to the teacher’s policy may not be optimal for the learner’s view. The paper assumes that the teacher has to not only provide demonstrations but also update the learner’s features. Assuming that the learner’s feature is a linear transformation of the teacher’s feature, this paper introduces a new measure “teaching risk” which can serve as a performance gap between the teacher and the learner given their different views. This paper further proposes an algorithm that greedily updates the feature of the learner and its policy to minimize the teaching risk. This algorithm outperforms two heuristic algorithms on NxN grid world domain. [Pros] - Introduces a new and interesting IRL problem. - Proposes a new performance measure for this problem with a good theoretical result. - Proposes an algorithm that performs better than heuristic approaches. [Cons] - The proposed approach and the experimental result are limited to the linear setting. 2. Quality - This paper introduces an interesting IRL problem with good motivating examples. - The proposed measure “teaching risk” is interesting and reasonable with a nice theoretical backup. This also leads to an algorithm in this paper. - The theoretical/empirical results of this paper are based on the assumption that the learner’s feature is a linear transformation of the teacher. This paper could be much stronger if it showed a more general/practical algorithm and empirical result that works well on complex domains, though I think this is out of scope. - It would be good to discuss “Third-Person Imitation Learning” [Stadie et al.] and “Time-Contrastive Networks” [Sermanet et al.] in the related work, as they also deal with imitation learning problem where the teacher and the learner have different views. 3. Clarity - The paper is very well-written. 4. Originality - The proposed problem/theoretical result/algorithm are novel to my knowledge. 5. Significance - This type of IRL problem (e.g., 3rd-person imitation learning) is becoming important as it is hard to provide 1st-person demonstrations in the real-world applications (e.g., robotics). Although the proposed algorithm and the experiment are limited to the linear setup, I think a good theoretical understanding provided by this paper would be useful for building a more practical algorithm in the future.

Reviewer 3



The paper proposes a new metric, namely, the teaching risk, in context of an inverse reinforcement learning (IRL) problem. It argues that under a feature mismatch of the world views between the learner and the expert (in an IRL setting), the teaching risk (i.e. the maximum performance gap between the teacher and the learner) will be higher. By lowering and bounding the teaching risk guarantee that the learner is able to learn a near-optimal policy under incomplete world view. The efficacy of the approach is tested in a grid-world setting where features for the IRL algorithm are selected based on the teacher risk metric (where the learner is provided with features which reduce its teaching risk by the teacher, learner's world view is updated, and a policy is learnt using the features by imitating learner's view of teacher's policy) and compared with randomly selected features and features based on greedily maximizing performance. The paper is well motivated but but the experimental evaluations are not convincing. The results provided are averaged over 30 experiments. It would be interesting to see the variance of the relative performance rather than just the mean. As discussed in the paper, one of the advantages of the TRGreedy feature selection is that it can find if additional sensors are needed for optimal performance. It would be interesting to see this tested in an actual scenario. Additionally, the paper should also discuss if the method can be implemented in a real world scenario or a grid world simulator[1-3]. Minor corrections: Confusion in Line 68:70 and 105:107 Figure 3 should be explained in more detail Typo in line 9: "guarantee" [1] Kolve, E., Mottaghi, R., Gordon, D., Zhu, Y., Gupta, A., & Farhadi, A. (2017). AI2-THOR: An interactive 3d environment for visual AI. arXiv preprint arXiv:1712.05474. [2] Mo, K., Li, H., Lin, Z., & Lee, J. Y. (2018). The AdobeIndoorNav Dataset: Towards Deep Reinforcement Learning based Real-world Indoor Robot Visual Navigation. arXiv preprint arXiv:1802.08824. [3] Savva, M., Chang, A. X., Dosovitskiy, A., Funkhouser, T., & Koltun, V. (2017). MINOS: Multimodal indoor simulator for navigation in complex environments. arXiv preprint arXiv:1712.03931. Update: I agree with the other reviewers that the problem formulation is novel but the discussion of experimental results are limited. I have increased my score to borderline accept assuming the authors make the changes discussed for the revised version in the author feedback.